# Effectiveness of *Aedes*-borne infectious disease control in Latin America and the Caribbean region: A scoping review

**Vaitiare Mulderij-Jansen**[1,2,3]*, **Prachi Pundir**[4], **Maria E. Grillet**[5], **Theophilus Lakiang**[6], **Izzy Gerstenbluth**[3,7], **Ashley Duits**[8,9,10], **Adriana Tami**[1], **Ajay Bailey**[2]

**1** Department of Medical Microbiology and Infection Prevention, University of Groningen, University Medical Center Groningen, Groningen, The Netherlands, **2** Faculty of Geosciences, Department of Human Geography and Spatial Planning, International Development Studies, Utrecht University, Utrecht, Netherlands, **3** Department of Epidemiology, Curaçao Biomedical & Health Research Institute, Willemstad, Curaçao, **4** George Institute for Global Health, New Delhi, India, **5** Facultad de Ciencias, Instituto de Zoología y Ecología Tropical, Universidad Central de Venezuela, Caracas, Venezuela, **6** Bridge Medicals Consulting Private Limited, Delhi, India, **7** Epidemiology and Research Unit, Ministry of Health Environment and Nature of Curaçao, Willemstad, Curaçao, **8** Red Cross Blood Bank Foundation, Willemstad, Curaçao, **9** Department of Immunology, Curaçao Biomedical & Health Research Institute, Willemstad, Curaçao, **10** Institute for Medical Education, University Medical Center Groningen, Groningen, The Netherlands

* v.i.c.jansen@umcg.nl

**Data Availability Statement:** All relevant data are within the paper and its Supporting information files.

## Abstract

### Background

*Aedes aegypti* and *Aedes albopictus* are primary vectors of emerging or re-emerging arboviruses that threaten public health worldwide. Many efforts have been made to develop interventions to control these *Aedes* species populations. Still, countries in the Latin America and the Caribbean (LAC) region struggle to create/design/develop sustainable and effective control strategies. This scoping review synthesises evidence concerning the effectiveness of *Ae. aegypti* and *Ae. albopictus* prevention and control interventions performed in LAC (2000–2021). The findings can be used to evaluate, compare and develop more effective control strategies.

### Methodology

The review is based on the methodology by Joanna Briggs Institute for conducting a scoping review. The MEDLINE (via PubMed and Web of Science), Cochrane Library, Scopus, EMBASE and ScienceDirect databases were used to search for articles. Grey literature was searched from governmental and non-governmental organisation websites. Four reviewers independently screened all titles and abstracts and full-text of the articles using the Rayyan web application, based on pre-defined eligibility criteria.

### Results

A total of 122 publications were included in the review. Most studies focused on dengue virus infection and data on *Ae. aegypti*. Entomological data were mainly used to determine the intervention's effectiveness. An integrated control intervention was the most commonly

**Funding:** Funding was provided by the Netherlands Organisation for Scientific Research (NWO) for the project entitled "Public health impact of chronic chikungunya illness and performance/utilisation of the health care system in the face of arboviral (dengue, chikungunya, Zika) epidemics in Curacao" Acronym: ARBOCARIB (NWO grant ALWCA.2016.021). The funders had no role in study design, data collection and analysis, decision to publish or preparation of the article.

**Competing interests:** The authors have declared that no competing interests exist.

employed strategy in both regions. Biological control measures, environmental management, and health education campaigns on community participation achieved more sustainable results than an intervention where only a chemical control measure was used. Challenges to implementing interventions were insufficient financial support, resources, workforce, intersectoral collaboration and legislation.

## Conclusions

Based on the synthesised data, an integrated vector (*Aedes*) management focused on community participation seems to be the most effective approach to mitigate *Aedes*-borne infectious diseases. Maintaining the approach's effect remains challenging as it requires multisectoral and multi-disciplinary team engagement and active community participation. Future research needs to address the barriers to program implementation and maintenance as data on this topic is lacking.

## Introduction

Worldwide, there are approximately 3,500 registered mosquito species [1]. Only a small portion of these species carry and transmit pathogens to humans [2,3]. Arboviral (arthropod-borne viral) diseases account for more than 17% of infectious diseases worldwide, affecting millions of people [4]. The global incidence of dengue virus infection has increased 30-fold in the last decades [5]. Dengue virus infection is an important public health issue with increasing morbidity and mortality in Latin America and the Caribbean region (LAC) [6]. Besides the dengue virus, chikungunya and Zika viruses are also of concern. They have rapidly spread throughout the LAC region in the last eight years, resulting in epidemics and high levels of morbidity, creating an added burden on the region's health systems [7].

Mosquitoes of the *Aedes* genus are considered the most important vectors of the mentioned arthropod-borne viruses [8]. In the LAC, *Aedes (Ae.) aegypti* is the primary mosquito vector of dengue, chikungunya, and Zika viruses [9]. In addition, *Aedes albopictus* (the Asian tiger mosquito) has also been shown to be a competent vector of the above-reported viruses [10,11]. *Ae. albopictus* is of medical importance due to its aggressive daytime human-biting behaviour, ability to adapt to colder climates and live in artificial and natural containers close to humans, resulting in disease transmission in new geographical areas. Since *Ae. albopictus* is also present in the LAC, countries in this region must consider this mosquito species' possible implications for transmitting viruses [12,13].

In the last decades, efforts have been made to develop vaccines, drugs, and mosquito control interventions to prevent and control diseases transmitted by *Ae. aegypti* and *Ae. albopictus* (*Aedes*-borne infectious diseases [ABIDs]). A licensed dengue vaccine is available; however, it is not widely used due to safety concerns [14]. For chikungunya and Zika virus infections, *Aedes* control is currently the only method available to prevent and control transmission. *Aedes* control aims to limit the transmission of pathogens by reducing or eliminating human contact with the mosquito.

The World Health Organization (WHO) recommends using an Integrated Vector Management (IVM) program to manage mosquito populations [15]. The characteristic features of the IVM program include (i) selection of methods based on knowledge of local vector biology, disease transmission and morbidity; (ii) utilisation of a range of interventions (e.g., biological,

chemical control measures, community mobilisation), often in combination and synergistically; (iii) collaboration within the health sector and with other public and private sectors; (iv) engagement with local communities and stakeholders; (v) a public health regulatory and legislative framework; (vi) rational use of insecticides; and (vii) good management practices. Ideally, the proper implementation of an IVM program could help countries deal with ABIDs. However, many countries, including countries in the LAC region, face challenges such as insufficient resources, collaboration within the health sector and with the public and private sector, workforce, training, and/or issues with immigration or cross-border transmission, which obstruct the development and implementation of the IVM program [16,17]. Also, social-cultural, environmental, and climatic parameters (e.g., urbanisation, building environment, and climate change) limit the scope of surveillance and entomological control [18]. On top of the mentioned challenges, other emerging infectious disease pandemics/ epidemics, such as the (COVID-19) SARS-CoV-2 pandemic, continues to impose substantial stress on health care systems, further jeopardising *Aedes* control interventions.

Previous outbreaks of ABIDs have shown the weakness in managing infectious disease spread in the LAC region [17,19]. Based on the history of *Ae. aegypti* and *Ae. albopictus*, it is undeniable that other ABIDs (e.g., Mayaro, Yellow fever, Venezuelan Equine Encephalitis and West Nile virus infections) represent emerging threats favoured by climate change, globalisation, and travel/ trade [20–22]. The danger of the emerging or re-emerging of diseases transmitted by *Ae. aegypti* and *Ae. albopictus* requires us to learn from our experiences. Also, we need to work towards more sustainable integrated *Aedes* control strategies, taking the current challenges of countries in the LAC region into account.

There are recent comprehensive reviews about interventions to control *Ae. aegypti* [23]. However, to our knowledge, reviews covering more than one *Aedes* species and ABID are lacking in scientific literature. Therefore, our review attempts to synthesise the evidence (scholarly journal articles and grey literature) regarding the effectiveness of *Ae. aegypti* and *Ae. albopictus* prevention and control interventions (against dengue, chikungunya and Zika virus) performed in LAC in the last twenty-one years (2000–2021). Furthermore, the reported challenges, lessons learned and recommendations to deal with the practical challenges in applying *Aedes* control interventions will be documented in the review. Countries in the LAC region can use the evidence generated from this article to (i) evaluate their IVM program holistically, (ii) compare their *Ae. aegypti* and *Ae. albopictus prevention and* control strategies with neighbouring countries, and (iii) apply lessons learned to develop more sustainable *Aedes* control approach to prevent and manage ABIDs in the era of COVID-19 pandemic and beyond.

## Materials and methods

This scoping review is based on the framework proposed by Arksey and O'Malley, which has been further developed by Levac *et al.* and the methods by Joanna Briggs Institute (JBI) [24–26]. The JBI framework recommends organising the review process in at least five stages: (i) identifying the research question, (ii) identifying relevant studies, (iii) study selection, (iv) charting the data, and (v) collating, summarising and reporting the results. The scoping review also adheres to the Preferred Reporting Items for Systematic reviews and Meta-Analyses extension for Scoping Reviews (PRISMA-ScR) checklist (S1 Table) [27]. The protocol of this scoping review has been registered in Open Science Framework (OSF), link: https://osf.io/dkzht/wiki/home/.

The eligibility criteria for the review were based on the 'Population-Intervention-Comparison-Outcome' (PICO) framework and not the PCC (Population-Concept-Context) framework

suggested by JBI. The PICO framework was chosen because it leads to comprehensive search strategies and yields more precise results. The included studies followed the following requirements:

## Population

The population included residents of the Caribbean islands and countries of the Latin American region of all age groups irrespective of gender, income, occupation, or other demographic characteristics. Each article was screened to check if the reported study population met our eligibility criteria. Articles that included hospitalised individuals with metabolic or fatal diseases/ terminal illnesses were excluded from the review.

## Intervention

The primary concept of this scoping review is *Ae. aegypti* and *Ae. albopictus* prevention and control interventions, including (i) insecticide-treated materials (e.g., curtains, nets or screens), (ii) usage of larvicides in breeding sites, (iii) usage of adulticides (e.g., outdoor fogging and indoor residual spraying), (iv) lethal oviposition trap-based mass interventions, (v) container management/reduction, (vi) health education, (vii) community engagement, (viii) media campaigns, (ix) biological control measures (e.g., usages of other living organisms, such as larvivorous fish and *Wolbachia*), (x) mosquito coils/ repellents, (xi) interventions focussed on behavioural change, advocacy (informed influence activities on policymakers from civil society), and (xii) integrated surveillance, epidemiological or entomological surveillance as part of a control program. All field experiments that were not population-based simulations or semi-field trials were excluded.

## Comparison

The comparison of the interventions for ABIDs was no intervention or the usual/ older intervention. The comparison groups or control groups could be a combination of interventions, such as biological control measures and community engagement in one group and the same interventions in two other groups individually. Such comparisons assisted in testing the effectiveness of a specific intervention. The studies that used surveillance data to assess intervention effectiveness and the pre-post study designs with no comparison groups were also included in this review. Cross-sectional studies were included when the collected data were compared with previously collected data (e.g., surveillance data).

## Context

This scoping review focused on countries located in the LAC region. The Latin American region consists of the following countries: Belize, Costa Rica, El Salvador, Guatemala, Honduras, Mexico, Nicaragua, and Panama in North and Central America. Argentina, Bolivia, Brazil, Chile, Colombia, Ecuador, French Guiana, Guyana, Paraguay, Peru, Suriname, Uruguay, and Venezuela are the countries in South America [28]. The Caribbean region consists of the following islands: Anegada, Anguilla, Antigua, Aruba, Bahamas, Barbados, Barbuda, Bonaire, Cayman Islands, Cuba, Curaçao, Dominica, Dominican Republic, Grenada, Grenadines, Guadeloupe, Haiti, Hispaniola, Jamaica, Jost Van Dyke, Martinique, Montserrat, Nevis, Puerto Rico, Saba, Saint Croix, Saint Martin, Saint Kitts, Sint Eustatius, Saint Barthélemy, Saint John, Saint Lucia, Saint Martin, Saint Thomas, Saint Vincent, Tortola, Trinidad and Tobago, Turks and Caicos Islands, Virgin Gorda, and Water Island [28].

## Outcome

The reviewers included publications that report the effect of the intervention (s) on (i) egg, larva, pupa or adult mosquito density, including entomological indexes such as; the Breteau index [(number of containers infested / total houses inspected) x 100], the container index [(number of containers infested / total containers inspected) x 100], the house index [(number of houses infested / total households) x 100], pupae per person index [number of pupae per household population] and pupae per Hectare index [number of pupae per household area], (ii) ABID incidence, (iii) knowledge, attitude, and practice (KAP), and (iv) environmental adaptations such as the reduction of mosquito breeding sites.

## Time-frame, language and other criteria

This scoping review does not focus on intervention cost, but studies containing information on both cost and effect of intervention were considered. The included articles were (i) published between January 1st 2000, and May 28th 2021, and (ii) were written in English, Spanish and Portuguese. An article was excluded if it (i) contained only entomological surveillance, epidemiological or prevalence data without a link to *Ae. aegypti* and *Ae. albopictus* prevention and control strategies; (ii) outcome of interest not reported, (iii) not available in full-text; and (iv) written in French. Publications from French-speaking islands/countries were included if the publication was written in English, Spanish or Portuguese. The reviewers did not exclude an article based on the year of data collection; all publications starting from the year 2000 were included.

## Electronic databases and other sources

The MEDLINE (via PubMed and Web of Science), Cochrane Library, Scopus, EMBASE and ScienceDirect databases were searched for articles. Additionally, Google Scholar and the Google search engine searched for the first 100 relevant results. Grey literature regarding *Ae. aegypti* and *Ae. albopictus* prevention and control interventions were searched on the governmental and non-governmental organisation websites such as the WHO library database, UNICEF, and Latin American and Caribbean Health Sciences Literature.

## Study design

This scoping review shows evidence from primary studies or journal articles or organisational reports, or dissertations (e.g., cross-sectional studies- analytical, case-control studies, cohort studies, randomised and non-randomised controlled trials, impact evaluations, qualitative and mixed-methods studies). We decided not to include systematic or scoping or narrative reviews and meta-analyses because we were interested in the singular study findings. All such 'reviews' reference lists were searched for additional primary studies on the research topic during the full-text screening stage. Conference proceedings, letters to the editor, editorials, and commentaries were excluded.

## Search

Two team members (PP and VJ) experienced in searching databases designed the search strategy (S2 Table). Keywords were identified based on three domains; (i) mosquito-borne infectious diseases, (ii) location, and (iii) *Ae. aegypti* and *Ae. albopictus* prevention and control interventions using Medical Subject Headings [MeSH], existing literature reviews, and subject experts' opinions. The keywords were combined with appropriate boolean operators to search for articles in electronic databases, and similar keywords were used for grey literature.

Proximity operators, truncation and wildcards were used for keywords to increase the sensitivity of the search. The search was initially conducted on PubMed and then tailored to other databases. PRESS checklist was used to review the search strategy's quality [29].

## Selection of studies

The compiled search results were de-duplicated using EndNote 20 and exported to rayyan.ai to screen blinded [30]. The reviewers stored the articles and other data files using cloud storage technology. The selection of studies was performed by four reviewers independently (VJ, MG, PP, TL), and disagreements were resolved in discussion with the team. The articles were screened twice (i) in the title and abstract screening stage and (ii) in the full-text screening stage. The selection process is presented in the PRISMA flow chart [31]. Three reviewers (VJ, PP, TL) piloted data extraction on a sample of the included studies (5% of the complete list of retrieved studies) to ensure that the data extraction technique was consistently applied. The data extraction was conducted by three reviewers (VJ, PP, TL) using a charting table. Data were extracted for the following variables: study identifiers, locations, study design, methods, demography information, intervention, comparison, outcome, challenges and recommendations.

A second reviewer (VJ) cross-checked each entry. The reviewers discussed doubts until consensus was reached or consulted a fourth reviewer to resolve the disagreements.

## Data analysis

The data were analysed using Microsoft Excel using basic formulae and data for the figure. Frequencies and percentages were calculated for the variables of interest for summarising the data, and a 'characteristics of included studies' table was prepared from the data charting sheet. Advanced analysis was not performed, and the quality assessment of the included studies was not carried out as both are not processes suggested by the JBI guidelines [26].

## Results

The results are presented here by the following key themes: (i) characteristics of the included publications, (ii) differences and similarities between regions, (iii) interventions, (iv) challenges and lessons learned, and (v) strategies employed or recommended to improve the effectiveness of *Aedes* prevention and control measures.

## Characteristics of the included publications

A total of 11,222 (11,118 articles from databases and 104 articles from other sources, e.g., google or reference list search) articles were identified by the reviewers in the initial search. After eliminating duplicates, the remaining articles were screened by reading their title and abstract. One hundred twenty-eight articles from databases and 101 articles from other sources were qualified and further screened by reading the full-text. Eighteen of the 128 articles from the databases and 89 articles from other sources were excluded based on the exclusion criteria. A total of 122 articles were included in the scoping review. Fig 1 shows a flow diagram of the selection process.

Of these 122 articles, 90 were from the Latin America region [32–121], and 31 [122–152] were from the Caribbean region. One publication contained information from both regions [153]. The characteristics of the included studies are presented in S3–S5 Tables. Most studies from the Latin America region were from Brazil (n = 30) [35–64], Mexico (n = 21) [90–109,120] and Colombia (n = 18) [65–82] (Table 1). Most studies from the Caribbean region

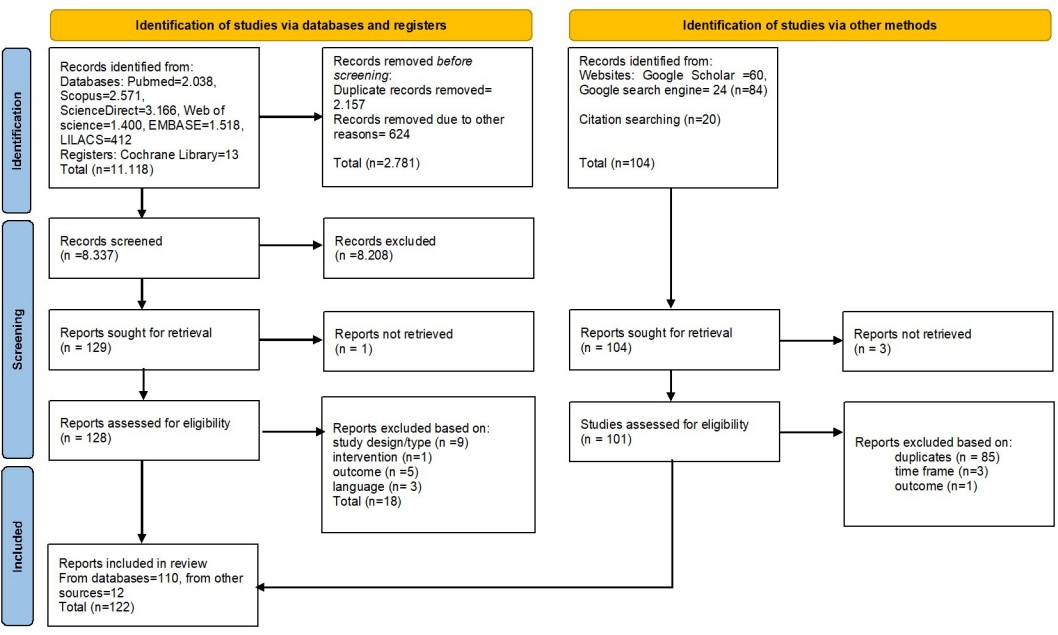

**Fig 1. PRISMA 2020 flow diagram.**

were from Cuba (n = 16) [122–137], Puerto Rico (n = 11) [139–149], and Trinidad (n = 2) [150,151] (Table 1). Basic characteristics of the included studies are presented in Table 1, and the number of publications for each year (2000–2021) is illustrated in Fig 2.

## Differences and similarities between the Caribbean and the Latin America region

Most of Latin America and the Caribbean region's publications have focused on the dengue virus infection, urban areas, and *Ae. aegypti* mosquitoes (Table 1). The *Ae. albopictus* mosquito has received more attention in Latin America than in the Caribbean. Chemical control measures were often applied in both areas compared to biological control measures. Mostly entomological data were used to determine the intervention's effectiveness, and it was more often used in Latin America than in the Caribbean region (Table 1). Health education campaigns were the most commonest employed (single) intervention to control ABIDs in both regions.

## Interventions

**Application of larvicide in mosquito breeding sites.** A study conducted in Brazil assessed the effect of a conjugate of *lysinibacillus sphaericus (Lsp)* and *Bacillus thuringiensis var. israelensis (Bti)* on *Aedes* eggs and adult mosquito populations [60]. No significant reduction in the *Ae. aegypti* adult population was proven, but a significant reduction in egg density was observed in the second year compared to the first year. In Puerto Rico, *Bti* was applied at a rate of 500 grams/ Hectare using vehicle-mounted aqueous wide-area larvicide spray applications [149]. This study found that *Bti* was successfully deposited into jars in both open and covered locations. After the intervention, differences in ovitrap densities were observed between the intervention and control group resulting in 62% (P = 0.0001) and 28% (P < 0.0001) reductions in adult female *Ae. aegypti* mosquitoes [149]. In Colombia, treating street catch basin with 2 grams of pyriproxyfen (approx. 0.05 mg/mL) was associated with a

**Table 1. Characteristics of the included studies.**

| | Caribbean n (%) (n = 31) | Latin America n (%) (n = 90) |
|---|---|---|
| **Type of Publication** | | |
| Journal article | 30 (96.8) | 88 (97.8) |
| Report | 1 (3.2) | 0 |
| Thesis/ dissertation | 0 | 2 (2.2) |
| **Language** | | |
| English | 26 (83.9) | 66 (73.3) |
| Spanish | 5 (16.1) | 16 (17.8) |
| Portuguese | 0 | 8 (8.9) |
| **Country (Top 3)** | | |
| Brazil | 0 | 30 (33.3) |
| Mexico | 0 | 21 (23.3) |
| Colombia | 0 | 18 (20.0) |
| Cuba | 16 (51.6) | 0 |
| Puerto Rico | 11 (35.5) | 0 |
| Trinidad | 2 (6.4) | 0 |
| **Setting** | | |
| Urban | 10 (32.2) | 41 (45.6) |
| Rural | 0 | 4 (4.4) |
| Both | 3 (9.7) | 7 (7.8) |
| Not mentioned | 18 (58.1) | 38 (42.2) |
| **Study designs** | | |
| Randomized controlled trial | 0 | 5 (5.6) |
| Non-randomized controlled trial | 15 (48.4) | 29 (32.2) |
| Cluster randomized trial | 7 (22.6) | 24 (26.7) |
| Pre-post (before-after) study | 6 (19.4) | 17 (18.9) |
| Mixed method study | 2 (6.4) | 2 (2.2) |
| Record surveillance | 1 (3.2) | 9 (10.0) |
| Cross-sectional study | 0 | 3 (3.3) |
| Other* | 0 | 1 (1.1) |
| **Demographics mentioned** | | |
| Yes | 6 (19.4) | 33 (36.7) |
| No | 25 (80.6) | 57 (63.3) |
| **Type of disease** | | |
| Dengue | 22 (71.0) | 75 (83.3) |
| Zika | 1 (3.2) | 0 |
| Chikungunya | 1 (3.2) | 0 |
| Other (combination) | 7 (22.6) | 15 (16.7) |
| **Type of mosquito** | | |
| *Ae. aegypti* | 29 (93.5) | 74 (82.2) |
| *Ae. aegypti* and *Ae. albopictus* | 2 (6.5) | 13 (14.4) |
| *Aedes* | 0 | 2 (2.2) |
| Not mentioned | 0 | 1 (1.1) |
| **Type of intervention (s)** | | |
| A. Application of larvicide (biological and chemical)** | 2 (6.5) | 4 (4.4) |
| B. Adulticiding (spraying of insecticide indoor/outdoor) | 0 | 10 (11.1) |
| C. Biological control (usage of larvivorous fish and *Wolbachia*) | 0 | 3 (3.3) |

*(Continued)*

**Table 1.** (Continued)

|  | Caribbean n (%) (n = 31) | Latin America n (%) (n = 90) |
|---|---|---|
| D. Environmental management (removal or covering of breeding sites, and usage of insecticide-treated curtains/screens) | 2 (6.5) | 14 (15.6) |
| E. Traps (including lethal traps for immature forms and adult mosquitoes) | 4 (12.9) | 7 (7.8) |
| F. Genetically modified mosquitoes | 0 | 1 (1.1) |
| G.Health education and community mobilisation | 5 (16.1) | 17 (18.9) |
| H. Integrated approach*** | 16 (51.6) | 26 (28.9) |
| A&B | 1 (3.2) | 0 |
| A&D | 0 | 3 (3.3) |
| B&D | 1 (3.2) | 0 |
| B&G | 0 | 1 (1.1) |
| D&G | 0 | 4 (4.4) |
| **Type of outcomes** |  |  |
| A. Entomological data (e.g., survey of mature/ immature mosquitoes) | 9 (29.0) | 40 (44.4) |
| B. Incidence of mosquito-borne infectious diseases | 1 (3.2) | 2 (2.2) |
| C. Knowledge-Attitude-Awareness-practices-perceptions | 2 (6.5) | 4 (4.4) |
| A&B | 4 (12.9) | 10 (11.1) |
| A&C | 5 (16.1) | 12 (13.3) |
| Other**** | 10 (32.3) | 22 (24.4) |

Note: The reviewers included one report documenting interventions performed in countries located in both Latin America and the Caribbean region. This report (USAID, 2019) is written in English, focusing mainly on Zika, but dengue was also mentioned. The following interventions were performed: Biological and chemical control measures, environmental management, usage of traps and health education. The outcomes were: Entomological data, removal of mosquito breeding sites, and improvement of knowledge, attitude and practices. The study designs were classified as non-randomized control trials.

Demographic information: Study participants' age, gender, and income or employment status.

* Economic modelling assessment.

** Application of larvicide (biological and chemical): *Bacillus thuringiensis var. israelensis*, Diflubenzuron, Novaluron, Spinosad, Temephos and Pyriproxyfen.

*** Integrated approach refers to a control strategy with more than three components (e.g., health education, environmental management and application of larvicide.

**** Type of outcome: More than three types of outcomes (e.g., self-reported dengue symptoms and anti-dengue IgM seropositivity rates combined with entomological parameters) or a type of outcome that has not been categorised, including removal of mosquito breeding sites, community participation, maintenance of activities through capacity building, Disability-adjusted life years (DALYs) and cost of interventions.

reduction in the incidence of dengue (rate ratio 0.19, 95% CI 0.12–0.30, P < 0.0001) [74]. Two studies conducted in Brazil used pyriproxyfen to control *Aedes* mosquitoes [52,56]. One study found that dissemination of pyriproxyfen dust-particles from dissemination stations led to a ten-fold decrease in adult mosquito emergence from sentinel breeding sites [52]. The second study also showed the beneficial effect of mosquito-disseminated pyriproxyfen. After the intervention, *Aedes* juvenile catch decreased by 79%–92%, and juvenile mortality increased from 2%–7% to 80%–90%. Also, the mean adult *Aedes* emergence fell from 1.077 per month (range 653–1,635) at baseline to 50.4 per month during the intervention (range 2–117) [56]. In Peru, mosquito breeding sites were treated with pyriproxyfen, but no significant effects (Breteau index: −1.05 with a 95% CI: −12.64–10.53 and pupae per person index: 0.365 with a

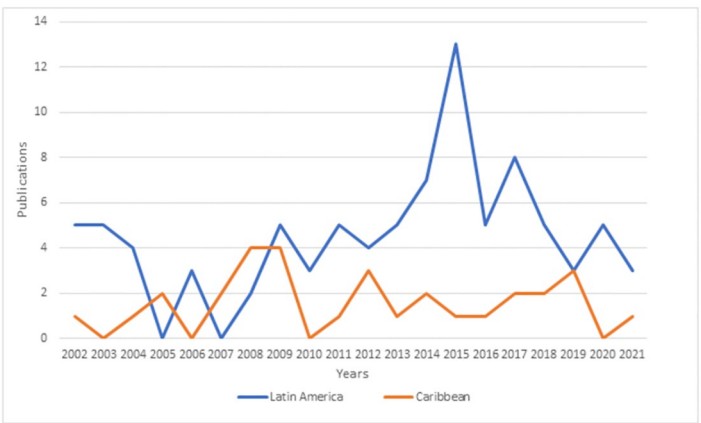

**Fig 2. The number of publications per year (2000–2021).**

CI: −0.030–0.760) were observed [120]. The observed beneficial effect could be due to the type of strategy (mosquito-disseminated insecticide) and not the kind of larvicide used.

A trial in Mexico assessed the effect of the application of spinosad (1 part per million [ppm], equivalent to one gram of active ingredient in one million millilitres of water), spinosad (5 ppm), 0.4 gram 1% temephos granules, and 50 µl Vectobac AS12 *(Bti)* on immature forms of *Aedes* mosquitoes located in car tires [92]. Both spinosad treatments (1 and 5 ppm) provided 6–8 weeks of effective control *of Ae. aegypti and Ae. albopictus* in the dry and the rainy season. *Bacillus thuringiensis var. israelensis* controlled *Aedes* larvae for one week. The duration of larvicidal activity of 1% temephos granules was intermediate between the spinosad treatments and *Bti*. The usage of temephos granules resulted in 4 weeks of complete control. A study in Trinidad that treated breeding sites with temephos (2–3 weeks before the onset of the rainy season) reported a significant decline of the *Ae. aegypti* population (P > 0.01), from a Breteau index of 19.0 to 6.0 and a pupae per person index of 1.23 to 0.35 [151].

The evidence shows the beneficial effect of larvicide application in mosquito breeding sites before and during the rainy season. The application of *Bti*, pyriproxyfen, spinosad and temephos in mosquito breeding sites led to a decline in the immature and adult *Aedes* mosquito populations. The usage of pyriproxyfen in breeding sites was associated with a decline in the incidence of dengue virus infection. However, it is essential to mention that the observed positive effect of pyriproxyfen is possibly caused by the method of disseminating the larvicide and not the larvicide itself.

**Spraying of insecticide (indoor/outdoor).**  Two studies that used insecticide spraying in Argentina [32,34] and Peru [114,115] were identified. After applying insecticide, these studies reported a reduction in *Aedes* mosquito larval indices and/ or adult density. In Brazil, the spraying of insecticide (malathion) using heavy equipment was associated with fewer cases of ABIDs compared to the control group (application of insecticide with portable equipment) [58]. Another study in Brazil found that ultra-low insecticide spraying avoided about 24% of symptomatic dengue cases in the study area throughout the 2015–2016 (December 1, 2015—June 30, 2016) dengue epidemic [62]. In Costa Rica, the application of ultra-low volume at the front door and in each room provided three weeks of significant control (P < 0.05) [83]. A study in Mexico that used metofluthrin emanators to reduce indoor adult *Ae. aegypti* abundance reported substantial reductions in abundance rate ratios for total *Ae. aegypti*, female abundance, and females that contained blood meals (2.5, 2.4, and 2.3-times fewer mosquitoes respectively; P < 0.001) [106].

Insecticide spraying led to a decline in *Aedes* mosquito density and incidence of ABIDs. The reported beneficial effect is up to seven months; thus, data on the longterm effect is lacking. The usage of heavy equipment led to more beneficial effects than portable equipment.

**Biological control.**   Two studies assessed the effect of usage of larvivorous fish in mosquito breeding sites in Latin America. In Brazil, usage of the fish *Betta splendens* led to a reduction in the presence of immature forms of *Ae. aegypti* mosquitoes (70.4% [January 2001] to 7.4% [January 2002] to 0.2% [December 2002]) [39]. In Mexico, larvivorous fish in breeding sites was associated with lower levels of recent dengue virus infection in children aged 3–9 years (OR 0.64; 95% CI 0.45–0.91) [102]. Another study in Brazil released *Wolbachia*-infected eggs as an intervention. The study reported a successful invasion and long-term (the post-release phase, spanning mid-January 2017 to December 2019) establishment of the bacterium across the study site [64]. The study results support the view that *Wolbachia*-infected mosquitoes have low susceptibility to dengue and Zika virus infections, with reduced viral replication and dissemination to humans [64]. The evidence above highlights biological control measures' beneficial and longterm effects on *Aedes* populations.

**Environmental management.**   Scheduled and periodic container washing led to a decline in *Aedes* (larvae) index in Peru [112]. In Venezuela, no significant difference in reducing the Breteau index (0.84 with a 95% CI: −8.94–10.62 and pupae per person index: −0.023 and a 95% CI: −0.749–0.703) was observed after covering drums with insecticide-treated nets [120]. A study in Brazil indicated that a long-term decrease in adult female population density was achieved only when water tanks and metal drums were covered with nylon net [45]. Another study conducted in Brazil that placed concrete in the bottom of storm drains indicated that after the intervention, water accumulated in 5 (9.6%) of the storm drains (P < 0.001), none (0.0%) had immature forms of *Aedes* species (P < 0.001), and 3 (5.8%) contained adults' mosquitoes (P = 0.039) [59]. Covering only the most productive breeding sites of *Ae. Aegypti* with netted lids also produced beneficial results [65]. The positive benefits of the prevention of water retention were also observed in a study conducted in Colombia [81].

Another study conducted in Colombia assessed the effectiveness of using long-lasting insecticide-treated curtains alone or in combination with container covers [76]. Long-lasting insecticide-treated curtains alone reduced the Breteau index from fourteen to six in the intervention group (the Breteau index in the control group was eight and reduced to five). A significant reduction in pupae per person index (P = 0.01) was observed when long-lasting insecticide-treated curtains were combined with the usage of container covers [76]. A similar study was conducted in Guatemala. Significant differences were observed when treated materials and other interventions targeting productive breeding sites (e.g., larviciding with temephos, elimination of breeding sites) were combined. The combined approach led to significant differences in reductions in the total number of pupae (P = 0.04), the house index (P = 0.01), pupae per person and the Breteau indices (P = 0.05) [86]. A study conducted in Mexico showed the long-term (more than two years) benefits of using insecticide-treated screens combined with treating the most productive breeding sites of *Ae. aegypti* [97]. In Mexico and Venezuela, a combined approach (using insecticide-treated curtains and treating water containers with pyriproxyfen chips or cover water containers) was also applied [108]. In both countries, entomological indices after the intervention were significantly lower than baseline. However, no significant difference between the control and the intervention group was observed due to the spillover effect (an indirect effect on a subject/ area not directly treated by the experiment).

In Mexico, insecticide-treated curtains usage was associated with fewer intradomicile dengue virus transmission [95], dengue virus-infected *Ae. aegypti* female mosquitoes [95,104], and *Ae. aegypti* mosquitoes' abundance [98,103,104,107]. In contrast with the benefits reported in the studies that used insecticide-treated curtains, a study conducted in Peru indicated that

despite the widespread use of treated curtains, individuals living in the intervention area were at greater risk of seroconverting to dengue virus (average seroconversion rate of 50.6 per 100 person-years CI: 29.9–71.9), while those in the control area had an average seroconversion rate of 37.4 per 100 person-years (CI: 15.2–51.7) [116]. A false sense of security may have caused the higher risk of dengue virus exposure observed in the mentioned study.

Two studies conducted in Uruguay distributed plastic bags for collecting unused small containers [117,118]. One study indicated that the vector densities in the intervention group, on average, increased less than those in the control group (from spring to autumn) after implementing the interventions (collection of small containers and covering of large containers). However, the difference was statistically not significant [117]. In the other study, the average pupae per person index decreased in the intervention group 11 times and in the control group four times (P < 0.05). Although the difference was not statistically significant, the container index, house index, and Breteau index decreased in the intervention group more than those in the control clusters [118].

One study in Cuba and Haiti used long-lasting insecticide-treated curtains or bed nets to control *Aedes* mosquitoes [135,138]. In Cuba, no effect of the insecticide-treated curtains on *Aedes* infestation levels (house index and Breteau index) was observed (study period 18 months) [135]. In contrast, the study in Haiti demonstrated significant differences between the intervention group and the control group. At one month post-intervention (usage of insecticide-treated bed nets), all entomological indices declined (house index in the intervention group declined with 6.7 (95% CI -10.6, -2.7; P < 0.01) and Breteau index reduced by 8.4 (95% CI -14.1, -2.6; P < 0.01) [138]. At five months, all indices remained low, and some were significantly lower than baseline in the control group [138]. Also, an IgM serosurvey demonstrated a 5.3% decrease (95% CI 5.0–25.5%, P < 0.01) in the number of IgM-positive individuals from baseline to the last survey.

In general, environmental management, especially combined approaches (e.g., using insecticide-treated screens and treating the most productive breeding sites), led to beneficial and even longterm effects ($\geq$ two years). However, it is crucial to be aware that the community perceptions/ participation and negligence of potential mosquito breeding sites can negatively affect the approach mentioned above's effectiveness.

**Traps.** In Latin America, eight studies that used traps as control measures were identified [38,42,43,50,54,67,72,73]. Five studies were conducted in Brazil, and three were performed in Colombia. One study conducted in Brazil found that sticky traps (MosquiTRAP) did not reduce the adult *Ae. aegypti* abundance and dengue infections were equally frequent in the intervention and the control group [54]. In contrast with the findings summarised above, other studies that combined a type of insecticide with traps demonstrated more beneficial results. For example, a study in Brazil that used traps with insecticide-treated ovistrip (impregnated with deltamethrin) reported a significant reduction in densities of *Ae. aegypti* for most comparators (P < 0.01), as shown by fewer positive containers and pupae/house at the intervention site compared to the control group [38]. Two other studies in Brazil used ovitraps with *Bti* for a massive collection of *Aedes* eggs [42,43]. Both studies indicated that massive egg collection by using ovitraps with *Bti* can affect the population density of *Aedes* mosquitoes. Two studies in Colombia also used ovitraps with *Bti*, and a significant reduction in entomological indices was observed [67,72]. The third study conducted in Colombia found that the ovitraps with the highest vector reduction combined deltamethrin/towel/10% hay infusion [73]. Another study performed in Brazil used BG-Sentinel traps for massive trapping of *Aedes* mosquitoes [50]. The findings of the mentioned study indicated that massive trapping with BG-Sentinel traps significantly reduced the abundance of adult female *Ae. aegypti* mosquitoes during the rainy season. However, no effect was observed in the dry season. Also, dengue

infections were lower in the area that used the traps compared to the control area; however, this was not statistically significant [50].

Five publications in the Caribbean region (all from Puerto Rico) reported using autocidal gravid ovitraps (AGO traps) as the primary control intervention [141–144,148]. Two studies reported significant reductions in mosquito density [141,142]. There were significant reductions in the captures of female *Ae. aegypti* (53–70%) in the study area. The presence of three to four AGO control traps per house (in 81% of the houses) prevented expected outbreaks of *Ae. aegypti* after rains. Mosquito captures in BG-Sentinel, and AGO traps were significantly and positively correlated, indicating that AGO traps are valuable and inexpensive mosquito surveillance devices [141]. One study reported a lower incidence of chikungunya virus infection in the intervention compared to the control group, resulting from tenfold lower mosquito densities in the intervention areas with AGO traps [143]. Two other studies also reported similar results [144,148].

The evidence on traps suggests that sticky traps are less effective than ovitraps (combined with a type of insecticide) and traps to capture adult mosquitoes. Ovitraps and traps to capture adult mosquitoes led to a significant reduction in entomological indices and a decline in the incidence of ABIDs.

**Genetically modified mosquitoes.**   One study released transgenic male *Ae. aegypti* mosquitoes with the OX513A line to assess the related changes in the distribution of infestation and abundance of *Ae. aegypti* populations six and eighteen months after the intervention in two areas in Brazil [57]. An average suppression of ± 70% of the wild population due to the release of transgenic males was observed in the mentioned study. In one of the areas, the mosquito population remained suppressed for 17 weeks, whereas in the other area, the suppression lasted 32 weeks [57]. The reported results highlight the benefits of using genetically modified mosquitoes in *Aedes* mosquito control.

**Health education and community mobilisation/participation.**   In Latin America, schools were a popular setting to provide health education concerning diseases transmitted by the *Aedes* mosquitoes discussed in this scoping review [36,70,77,79,80,88,89,93,96,100]. The health education interventions implemented at schools increased awareness of *Aedes* biological characteristics [36,96] and dengue prevention and control practices [70,88]. Also, infestation rates of immature forms of *Aedes* mosquitoes in schools were reduced [77,79,89]. A health education program implemented in Mexico also influenced the parents' behaviour [93]. According to the mentioned study, the entomological indices decreased significantly (P < 0.05) in houses in the intervention area, apparently because parents acted on the comments of the children and eliminated or monitored mosquito breeding sites [93]. A study that evaluated a health education campaign's sustainability (two years post-intervention) to prevent dengue in schools reported that the intervention's effects were not sustained. The intervention still affected the KAP of the schoolchildren; however, the results were not statistically significant [80].

Health education campaigns implemented among the community members also showed beneficial results in the Latin American region; (i) improvement of knowledge and reduction of mosquito breeding sites [37,48,75] and (ii) reduction of entomological indices [91]. A study that used a learning platform on mobile devices to improve KAP reported significant changes in attitudes and behaviour (P = 0.032) concerning actions to prevent arboviral diseases [61]. Another study that disseminated information via mobile phones suggested that repeated exposure to health information encourages householder's uptake of preventive measures against dengue [113]. One study that assessed the impact of mass-media communication campaigns indicated that although intervention coverage was adequate (59,4% of the population), the reach (people that could recall the content of the information) was low (22,3%). Also, no

association between the intervention and the presence of breeding sites was found [71]. Another study reported that mass communication campaigns influenced the population's KAP [136]. However, information gaps continue to exist, and actions beyond just providing information are required for better results. A study that combined mass communication campaigns with a school and museum-based educational program reported that exposure to the intervention was associated with increased knowledge about dengue. [139]. Also, the intervention was associated with an increased proportion of tires protected from rain and a decreased proportion of water storage containers positive for mosquito larvae [139].

A health education campaign based on an eco-health approach, focusing on community mobilisation, house inspection, and covering water containers with insecticide-treated aluminium covers showed an average decrease between 0.12 (-0.25–0.01) and 0.26 (-0.42 - -0.10) cases of dengue daily (1.82 cases per week or 7.8 cases per month or 95 cases per year) [82]. The benefits of health education campaigns focussing on community mobilisation and participation have been demonstrated in other study settings in the Latin American region [53,66,68,78,101,109,111]. Most studies on health education and community mobilisation/ participation in the Caribbean region were conducted in Cuba [122–125,127–129,131,132]. Health education and community mobilisation/ participation led to a decline in entomological indices [122–125,128,129,131], breeding sites [127], and behavioural change [128]. One study attributed the success of the intervention to community involvement in the vector control intervention [132]. The beneficial impact of health education focussing on community mobilisation was also observed in Puerto Rico [146].

The evidence concerning health education campaigns highlights the beneficial effect of health education on KAP of school children/ community members and entomological indices. However, the evidence also suggests that only education provision is not enough to control *Aedes* mosquitoes in the long run. Health education campaigns must include community participation/ mobilisation efforts to be successful. Furthermore, the government, including the health systems, must formalise mosquito control programs and health education campaigns through regulations to support interventions. Without regulations, it is hard to maintain program implementation.

**Integrated interventions.**   A study that evaluated the impact of the Brazilian national dengue plan (information campaigns, epidemiological surveillance and vector reduction interventions, e.g., usage of larvicide temephos and spraying of insecticide) indicated that the goals concerning the reduction of dengue incidence (50% reduction in dengue cases) and larval infestation (less than 1%) were not achieved in most municipalities (municipalities in the Southeast and Centre-West regions of Brazil) [44]. Another study that evaluated Brazil's national dengue control plan, in the municipality of Caruaru, attributed the program's lack of success to insufficient household coverage [46]. More positive results were achieved when the national program was extended (in two municipalities of Pernambuco -Brazil: Ipojuca and Santa Cruz) with the following interventions: ovitraps loaded with *Bti*, source elimination campaigns and indoor collections of adult mosquitoes using aspirators, targeting places considered highly important for virus transmission [49]. After two years of sustained control, a 90% decrease in egg density was registered at one study site. Data from another study site showed a sharp decline in the mosquito population [49]. The incorporation of the community mobilisation concept into the Brazilian national plan was linked with the reduction of the total number of dengue cases between 2009 and 2010 in Ipatinga city [55]. Combining the family health program with the national dengue control program can also provide beneficial results, as observed in a Brazilian study (São José do Rio Preto) [40]. Significant changes in KAP were achieved, and the house index was reduced (6.9% before and after 4.4%, with a significant difference, P = 0.040) [40].

A study in Mexico that combined epidemiological surveillance, environmental management through social mobilisation, and chemical control also demonstrated beneficial results [99]. Statistically significant differences (p < 0.001) in the relative abundance of *Ae. aegypti* larvae before and after applying larvicide temephos and removing water containers were reported [99]. For example, in La Paz, the average house index was reduced from 16–83% to 0–5% after the interventions for three years. Beneficial results were also observed in the positive container index and the Breteau index in La Paz and other areas (Cabo San Lucas and San Jose del Cabo), where the interventions were implemented [99]. The beneficial effects of combining interventions were also documented in another study conducted in Mexico [94]. Two years of implementing an integrated intervention reduced the peak of dengue cases recorded in the rainy season in Colima, Mexico [94]. However, more research is needed to determine the actual effect of the intervention since different factors (e.g., amount of rainfall) can influence the outcome of this study.

Other studies also demonstrated the benefits of combined interventions [33,41,51,63,84,85]. In Brazil, one study that combined the application of insecticide temephos, environmental management and health education campaigns did not observe positive results [35]. The lack of effect is possibly caused by negligence in eliminating breeding sites observed in the group that also applied temephos and the dilution of the insecticide in non-removable containers. A study in Nicaragua also did not demonstrate a positive effect of temephos combined with other interventions [110]. In the mentioned study, temephos exposure was not associated with a reduction in any entomological indices, and in some cases, temephos exposure was even associated with higher entomological outcomes [110].

In Cuba, an intervention combining the distribution of new ground-level water tanks and intensive use of an insecticide was conducted [126]. The container index decreased significantly from 0.7% to 0.1% one month after the intervention. Six months later, the mentioned index increased to 2.7% due to the uncovered new water container [126]. In Trinidad, an intervention combining the application of temephos in water containers, fenthion at walls of houses of suspected dengue cases and fogging using malathion failed to achieve the desired target of reducing mosquito densities to below the disease transmission threshold or possibly a Breteau index of five [150].

Another study conducted in Cuba implemented a community-based intervention focussing on (i) training a local task force, (ii) organising community working groups, (iii) enhancing collaboration between the government and the community, (iv) and creating a formal link with the routine vector control programme [130]. This intervention led to a lower attack rate of dengue fever (8.5 per 1000 inhabitants) in the intervention group compared to the control group (38.1 per 1000 inhabitants). A study in Puerto Rico reported a reduction in mosquito density after implementing an intervention that consisted of environmental management, usage of larvicide (Altosid Pro-G), and placement of three AGO traps in the backyards of houses [145]. Density changed from 27.7 mosquitoes/trap/week before to 2.1 after intervention (92.4% reduction), whereas after treating the original control area (cross-over), density changed from 22.4 to 3.5 (84.3% reduction) [145]. The mentioned results were confirmed by another publication [147].

The evidence presented above highlights the importance of combining interventions to mitigate ABIDs. An integrated approach can reduce entomological indices and the incidence of ABIDs. However, an integrated approach can also fail to achieve the desired effect if the community is not involved, mosquito breeding sites are not properly removed or treated, and insecticide is ineffective.

## Challenges and lessons learned

Another objective of this scoping review was to describe the challenges and lessons learned reported in the publications. The challenges and lessons learned are presented here by the following key themes: (i) knowledge concerning *Aedes* control and ecological adaptability, (ii) resources and capacity, and (ii) infrastructure and context of the community.

**Knowledge concerning *Aedes* control and ecological adaptability.**   The first step towards *Aedes* control is understanding the mosquitoes' basic biological and ecological characteristics. In most included studies, lack of community involvement, motivation, and community and health workers' understanding of *Aedes's* control immediate and long-term impact have been reported as significant barriers to successful program implementation [37,68,126]. Deciding on the methods for controlling calls for a deeper understanding of the operational aspects of mosquito control (the local conditions) and the ecological adaptability of the *Aedes* mosquitoes. For example, one study observed the evolution of mosquito resistance due to intense insecticide application [51]. Nowadays, *Ae. aegypti* can also use cryptic and non-clear aquatic habitats as oviposition sites. This change in the ecology of the mosquito can affect the results of the traditional surveillance and control methods. Furthermore, it is essential to understand community perceptions, control behaviour, and preferred control interventions [54]. For example, using a chemical substance such as temephos was linked with an increased risk of dengue infection, resulting from a false sense of security engendered by knowledge of pesticides in water storage [109].

**Resources and capacity.**   Insufficient resources, specifically human resources, to perform mosquito control is also a potential contributing factor that hinders mosquito control intervention's success [46]. For a sustainable intervention, the key is to have the political will to provide adequate and uninterrupted supplies [80]. A study in Guatemala reported financial constraints and scarcity of vehicles, fuel, and personnel to obstruct the sustainability of interventions [87]. Another study indicated that providing adequate basic utility services (e.g., garbage collection) to the communities could have a major bearing on the sustainability of the community-based mosquito control intervention, facilitating the successful implementation of the *Aedes* control program [124]. Proper public management (e.g., development and approval of health regulations and public health interventions, intersectoral coordination and creation of linkages between organisations to create a platform to negotiate solutions) is an essential factor that is required to achieve sustainable results [124]. Targeting long-term behaviour change among the community members is imperative for sustainability. Actively involving household members in basic mosquito control measures is a good start to avoid dependency on the program implementer and ensure program sustainability [117]. Mosquito control is a shared responsibility and requires coordination and support from all relevant sectors [46,86]. Programs need to be evaluated, and networks among the stakeholders need to be maintained.

**Infrastructure and context of the community.**   Important factors that might influence the effectiveness of *Aedes* mosquito control are the housing and surrounding conditions, methods of waste disposal, and socioeconomic status of the community [58]. Hence, detailed assessments of all public services, such as waste management, sewerage systems, etc., should be made to inform entomological surveys of *Aedes* mosquitoes [140]. The success of *Aedes* control intervention is also dependent on the socio and geopolitical condition of the community. In areas where community organisations are fragile or non-existent or in a conflicted community with a history of violence and unrest, social participation is slim to none [53,109].

## Strategies employed or recommended to improve the effectiveness of *Aedes* control interventions

The last objective of this scoping review is to list the employed or recommended strategies to improve the effectiveness of *Aedes* control. The recommendations are listed below:

1. Community participation enhances the success of *Aedes* control interventions [78,85,111,118,122,123,146]. Schools are an important setting that needs to be considered when designing community participation strategies since beneficial results have been booked with health education campaigns employed at schools [36,49,79,100].

2. The use of a multi-sectoral approach or inter-sectorial integration [40,89,90,127,128,139], integrated *Aedes* control approaches [97,111,147], and prioritizing ABIDs (local govern-ment providing financial and administrative support) [121,122] were stated as common reasons for the success of an intervention.

3. The effectiveness of the intervention improved when qualified staff planned and imple-mented an intervention [46,68], and the community accepted the intervention technique and believed in its effectiveness [107].

4. Interventions were effective when health education messages were repeated throughout the intervention [113].

5. The health workforce should be trained and capable of identifying the most productive breeding sites [74]. The workforce should keep standard operating procedures for entomo-logical and epidemiological surveillance, and data should be consistent and follow stan-dardised measurement tools for inspection activities [68,128]. Furthermore, the workforce should assess the short term and long-term impact of the *Aedes* control program because these assessments facilitate program implementation in the long run and sustainability. To assess immediate outcomes, periodic monitoring and evaluation are recommended to inform what works and needs to be amended [68]. Having a list of program indicators could be beneficial for program implementers to quantify and draw inferences on the out-comes against the input, which can be helpful for the allocation of resources.

6. It is also important to consider conducting a cost-benefit analysis as the intervention pro-gresses, considering the direct cost of implementing the intervention and the potential con-sequences [91].

7. Lastly, the workforce should consider factors (e.g., meteorological conditions such as wind speed, direction, or temperature) contributing to differences in larval mortality related to application data [149]. Another factor that can invariably affect implementation if not counter for is the seasonal variation of pupal productivity [86]. Therefore, a baseline pupal survey should be repeated every season to take seasonal variations of *Aedes* populations into account when designing an intervention.

## Discussion

This scoping review synthesises evidence concerning the effectiveness of *Ae. aegypti* and *Ae. albopictus* prevention and control interventions performed in LAC in the last twenty-one years (2000–2021). To our knowledge, this review provides the most updated overview of evi-dence regarding the effectiveness of *Aedes* prevention and control interventions in the region mentioned above. Most of the data were from studies in Brazil, followed by Mexico and Colombia. Cuba has most of the publications in the Caribbean region, followed by Puerto Rico

and Trinidad. The synthesised data showed that combined interventions were more effective than a single approach. Another review supported our findings by reporting that it is unlikely that any single intervention will be fully effective against *Aedes* mosquitoes and encourages the development and usage of an IVM program to control the mentioned mosquito species effectively [154].

Although our scoping review has a broader scope than a previously published systematic review and meta-analysis on the effectiveness of interventions for the control of *Ae. aegypti* in the LAC region [155], the same conclusions concerning the beneficial effect of an IVM program are drawn. An IVM program has been pointed out as the most effective approach to reducing *Ae. aegypti* immature forms and adult mosquitoes [155,156]. The WHO and the PAHO insist that the IVM approach is the solution to mitigate ABID transmissions [15]. However, many factors such as insufficient funding, resources, workforce, and political priorities (e.g., prioritising SARS-CoV-2) obstruct the implementation and maintenance of the IVM program nowadays. Besides, the focus of many health systems is not on mosquito control since many countries face a range of more immediate problems. On one hand, it is easy to recommend developing and implementing an IVM program. On the other hand, maintaining a cycle of periodic implementation of the program is challenging—this leaves the door open for future epidemics.

The most important question that remains is, how can we sustain the effectiveness of the IVM program? One way to counter this is through routine monitoring and evaluation, building the capacity of the community, building a network of relevant stakeholders (local and international), and taking the local challenges of the health system and the community into account when designing the IVM program. However, all these would require multisectoral and multi-disciplinary team engagement and active community participation. An IVM without community participation and the political commitment of the governments is not sustainable. [155,157].

The effectiveness of the interventions included in this review was reduced by the following factors, the spillover effect [76,108,138], small sample size [117], providing educational strategies to deploy control measures in the control group [68], parallel government campaigns [76,85,86,88,129,137], insecticide resistance [35,110,150] or media freedom to change or stop the distribution of information [71]. Also, most interventions were followed for a short period, which led to limited evidence on the sustainability of the employed interventions. Future research should focus on *Ae. Albopictus* and its active role in transmitting ABIDs. Also, researchers might consider studying the impact of an IVM approach on various ABIDs (e.g., dengue and Zika combined) since this evidence is also missing in the scientific literature. Furthermore, more research and dialogue on the barriers of an IVM program are needed to work toward more sustainable programs. This scoping review also aimed to synthesise evidence on how other countries in the LAC region dealt with their challenges, but this information was not available in all the included publications. Therefore, we recommend future research to address this topic to provide more practical advice to countries struggling with their *Aedes* control approaches. Lastly, we encourage countries with more advanced techniques and qualified workforce to support countries with less developed health systems.

## Limitations and strengths

Since the quality of the included studies was not assessed in this scoping review, the included studies' results need to be interpreted with caution. Quality assessment is essential for systematic reviews and meta-analysis, but it is not required for scoping reviews [26]. The main strength of this scoping review is the extensive literature, grey literature and reference list

search that was conducted to find relevant publications. Another strength is the methodology used; each phase in data search, screening and extraction was piloted and cross-checked to reduce the chance of bias.

## Conclusion

Prolonged community participation is key to sustaining the effectiveness of *Aedes* control interventions. The major conclusion of the synthesised data was the higher effectiveness of the integrated *Aedes* control approach over single strategies. Although it is known that an integrated approach in combination with community participation can be effective, many barriers still obstruct the development and implementation of such an approach, jeopardising the health system's preparedness and performance. Future research and recommendations need to focus more on IVM program implementation and maintenance barriers since this information is lacking in scientific literature.

## Supporting information

**S1 Table. Preferred Reporting Items for Systematic reviews and Meta-Analyses extension for Scoping Reviews (PRISMA-ScR) Checklist.**
(DOCX)

**S2 Table. Listed essential concepts and keywords.**
(DOCX)

**S3 Table. Characteristics of studies identified in the Latin America region about *Ae. aegypti* and *Ae. albopictus* prevention and control interventions.**
(DOCX)

**S4 Table. Characteristics of studies identified in the Caribbean region about *Ae. aegypti* and *Ae. albopictus* prevention and control intervention.**
(DOCX)

**S5 Table. Characteristics of studies identified in the Latin America and Caribbean region about *Ae. aegypti* and *Ae. albopictus* prevention and control interventions.**
(DOCX)

## Acknowledgments

We want to thank both the editors and reviewers for their insightful comments on the article.

## Author Contributions

**Conceptualization:** Vaitiare Mulderij-Jansen, Prachi Pundir, Maria E. Grillet, Ajay Bailey.

**Formal analysis:** Vaitiare Mulderij-Jansen, Prachi Pundir, Theophilus Lakiang.

**Funding acquisition:** Adriana Tami, Ajay Bailey.

**Methodology:** Vaitiare Mulderij-Jansen, Prachi Pundir.

**Project administration:** Adriana Tami.

**Supervision:** Maria E. Grillet, Izzy Gerstenbluth, Ashley Duits, Adriana Tami, Ajay Bailey.

**Visualization:** Vaitiare Mulderij-Jansen.

**Writing – original draft:** Vaitiare Mulderij-Jansen.

**Writing – review & editing:** Prachi Pundir, Maria E. Grillet, Theophilus Lakiang, Izzy Gerstenbluth, Ashley Duits, Adriana Tami, Ajay Bailey.

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
