## [Decision Letter · Decision Letter 0]

24 Aug 2022

PONE-D-22-16967Effectiveness of Aedes-borne infectious disease control in Latin America and the Caribbean region: a scoping reviewPLOS ONE

Dear Dr. Mulderij-Jansen,

Thank you for submitting your manuscript to PLOS ONE. After careful consideration, we feel that it has merit but does not fully meet PLOS ONE’s publication criteria as it currently stands. Therefore, we invite you to submit a revised version of the manuscript that addresses the points raised during the review process.

We look forward to receiving your revised manuscript.

Kind regards,

Rafael Maciel-de-Freitas

Academic Editor

PLOS ONE

Journal Requirements:

Reviewers' comments:

Reviewer's Responses to Questions

**Comments to the Author**

1. Is the manuscript technically sound, and do the data support the conclusions?

Reviewer #1: Yes

Reviewer #2: Yes

2. Has the statistical analysis been performed appropriately and rigorously? 

Reviewer #1: N/A

Reviewer #2: Yes

3. Have the authors made all data underlying the findings in their manuscript fully available?

Reviewer #1: Yes

Reviewer #2: Yes

4. Is the manuscript presented in an intelligible fashion and written in standard English?

Reviewer #1: Yes

Reviewer #2: Yes

5. Review Comments to the Author

Reviewer #1: This is a very usefull and wellcome review work. It surveys different Aedes control strategies, including approaches that are currently gaining ground, related to the growing importance of encouraging popular participation, extrapolating the merely biomedical interventions, more classically used.

However, I suggest, since the Introduction, to give more emphasis to the fact that the quality of the articles was not verified. Some conclusions of the original authors are reproduced here, but sometimes there is doubt as to whether good parameters and controls were used. Some examples have been added below.

The Introduction is adequate and updated, with justified arguments, but it is worth mentioning structural determinants, which go beyond the health sector and which limit the scope of surveillance and entomological control.

In the Results section, in the item 'Interventions', when referring to references 51 and 55, the text says that the use of pyriproxyfen '... led to a ten-fold decrease in adult mosquito emergence...'. It is necessary to make it clear here that the difference is the strategy, not the insecticide: this is the effect from the 'disseminating stations' (‘Mosquito-Disseminated Insecticide for Citywide Vector Control').

Still in the item ‘Interventions’, there are issues that are certainly not addressed in the analysis in general review studies of this type. When mentioning, for example, reference 91, which deals with the application of different larvicides, the stability and persistence of Bti formulations in the field, and the resistance of the vector population to insecticides, notably temephos, would be relevant parameters for the analysis of the results.

Regarding the last paragraph of the item 'Interventions', concluding about the material presented, I reinforce the importance of mentioning the disseminating stations, in this specific case, so that it does not seem that pyriproxyfen is more efficient than the other larvicides presented. The aspect at issue in articles 51 and 55 is not simply the product, but the way of disseminating it in the treated area.

Regarding the item 'Health education and community mobilization/participation' - considering that this control component has increasing relevance, a comment on the formalization, or not, of information on Aedes aegypti in schools in different countries would be worth mentioning. Perhaps in grey literature... Did the authors find something relevant, formal, in this sense?

About Reference 70, the text says that One study that assessed the impact of mass-media communication campaigns did not show beneficial results [70].' It seems a somewhat superficial conclusion, taking into account that, as mentioned in the Manuscript, the quality of articles has not been verified. In this specific case, what was the potential for publicizing the campaigns? Was it verified, for example, if the disclosure actually reached the affected people?

Still referring to the item ‘Health education and community mobilization/participation’, a comment: the number of articles from Cuba (121-124, 126-128, 130, 131) calls attention. The importance of popular participation in dengue prevention by means of Aedes control in this country is, in fact, already consolidated...

In fact, the importance of Cuba in relation to prevention actions, in general, is reiterated in the item 'Resources and capacity', in reference 123 (Another study indicated that providing adequate basic utility services (e.g., garbage collection) to the communities could have a major bearing on the sustainability of the community-based mosquito control intervention, facilitating the successful implementation of the Aedes control program [123].'). These papers signal not only the stimulus to popular participation, but also the counterpart of public management – an aspect that would deserve to be highlighted.

In some cases, throughout the manuscript, anecdotal reports are generalized, as if they were related to an entire country; it would be usefull to verify throughout the text. For example, in the item 'Integrated interventions', about reference 45: Another study that evaluated Brazil’s national dengue control plan attributed the program’s lack of success to insufficient household coverage [45].', it would be important to mention here that the article deals with the work in one municipality only.

Regarding reference 54: The incorporation of the community mobilisation concept into the Brazilian national plan was linked with the reduction of the total number of dengue cases between the years 2009 and 2010 [54].' - there is a difference between the title in Portuguese and English in the original article. The article in Portuguese refers to only one municipality, Ipatinga. In table S3, it corresponds to article 24. The table is ok, it alludes to Ipatinga. But the main text is confusing, there is room for the interpretation that the result obtained is national in scope.

Still in the item 'Integrated interventions', when there is mention of a study in Mexico (reference 93), the statement Dengue incidence decreased from 81.4% in 2010 to 79.1% in 2011 [93].', seems to be a vague conclusion: one point is that differences are too small; moreover, the fact that dengue does not occur homogeneously between years is not being taken into account. For example, the text in table S3 (it is article 63) says that: ‘The interventions reduced the peak of cases that had been recorded in the rainy season resulting from the transmission of dengue’ - that is, it seems that the text of the present Manuscript assumes that (a) there is a significant reduction in the number of cases in the period and that (b) this reduction is a function of the intervention. However, it is known that there are many other factors that may be contributing to this scenario, which could be at least cited, so as not to appear that this conclusion is being shared here.

The first mention of the possibility of insecticide inefficiency is in the final, concluding paragraph of the item ‘Integrated interventions’. It is worth dealing with this aspect, in a generic way, since the introduction, explaining that this variable exists, but it was not addressed in the bibliographic survey, neither taken into account in the analyses.

In the Discussion, in relation to the statement However, many factors such as insufficient funding, resources, workforce, and political priorities (e.g., prioritising SARS-CoV-2 over ABIDs) obstruct the implementation and maintenance of the IVM program nowadays.' – about 'SARS-CoV-2 over ABIDs': what do you mean 'political priority'? That was a contingency. It is not a question here of opposing one disease to another, but of ensuring a more comprehensive view of health. Finally, it is important to take into account that the focus is not the mosquito.

The Discussion refers to article 149 to exemplify resistance to insecticides as a factor in reducing the effectiveness of the interventions. This article corresponds to reference 29 of the list in table S4. However, there is no mention of resistance in the table, nor in the abstract of the article. In practice, the only occasion this article, 149, deals with resistance is in its Discussion section, when it cites two other references, both from 1998.

In the Conclusions, the manuscript states that The integrated Aedes control approach was more effective than a single strategy.'. Considering that there was no evaluation of data quality of the articles, as explained in the paragraph before this one in the main text (‘limitations and strengths’), it is not clear how this conclusion was reached. It is quite true that this conclusion is, at least intuitively, desired. But the parameters evaluated here do not allow such conclusions. On the other hand, it seems relevant (and could be emphasized) that this work contributes to systematize, classify, or quantify the different methodologies approached; it would be possible, for example, to present, even graphically, the relative contribution of different countries, or regions, to each type of approach. Another option is to make it clear that the higher effectiveness of an integrated approach over single strategies was the major conclusion of the analyzed articles - and not of the authors of this manuscript.

In the Acknowledgments, the 'thank you in advance' calls attention: We want to thank both the editors and reviewers for their insightful comments on the article.'

Minor comment: I did not systematically check the bibliographic references, in relation to questions of form. However, I noticed that reference 34 (Camargo Donalisio MR, Leite OF, Mayo RC, Pinheiro Alves MJC, de Souza A, Rangel O, et al. Use of Temephos for Control of Field Population of Aedes aegypti in Americana Sao Paulo, Brazil. 2002.') is incomplete. It remains to add: ‘Dengue Bulletin. 2002 Dec; 26: 173-177'.

Reviewer #2: Is a well written and important scoping review, my congratulations to the authors. This review give valuable information for control programas in Latin America and the Caribbean. In tha attached file, the authors can found somo suggestions.

6. PLOS authors have the option to publish the peer review history of their article (what does this mean?). If published, this will include your full peer review and any attached files.

Reviewer #1: No

Reviewer #2: **Yes: **Gabriel Parra-Henao

---

## [Author Response · Author response to Decision Letter 0]

15 Oct 2022

Review PONE-D-22-16967

Dear editor and reviewers,

We want to thank you for your insightful comments. Thank you for the time invested in our manuscript. Your valuable and insightful feedback led to improvements in the current version. We have carefully considered the comments and tried our best to address every one of them. We hope that the manuscript meets now your high standards. The authors welcome further constructive remarks if any. 

Below we provide a point-by-point response. All modifications in the manuscript have been highlighted in red (see file manuscript with track changes). 

Sincerely,

Drs. Vaitiare Mulderij-Jansen (PhD student)

Drs. Prachi Pundir

Prof. dr. Maria E. Grillet

Drs. Theophilus Lakiang

Drs. MD. Izzy Gerstenbluth 

Prof. dr. Ashley Duits

Dr. MD. Adriana Tami

Prof. dr. Ajay Bailey

Editor:

Please ensure that your manuscript meets PLOS ONE’s style requirements, including those for file naming.

Thank you for raising this point. We have used the PLOS ONE style templates to ensure that the manuscript meets PLOS ONE’s style requirements.

In your Data Availability statement, you have not specified where the minimal data set underlying the results described in your manuscript can be found. PLOS defines a study’s minimal data set as the underlying data used to reach the conclusions drawn in the manuscript and any additional data required to replicate the reported study findings in their entirety. All PLOS journals require that the minimal data set be made fully available. For more information about our data policy, please see http://journals.plos.org/plosone/s/data-availability.

Thanks for the remark. Each included study is cited. The data used in the studies are online (see published article). The following data availability statement has been added to the manuscript: “Data Availability: All relevant data are within the manuscript and its supporting information files.” (See page 31). We hope we have addressed this point correctly.

Thanks for raising this point. We have reviewed the reference list. One reference was added to the introduction, reference number 18. No papers that have been retracted were included in this scoping review. Also, we have checked if the names of the journals are abbreviated and if the date format is correct, especially in the case of articles published on websites.

Reviewers:

Reviewer #1: This is a very usefull and wellcome review work. It surveys different Aedes control strategies, including approaches that are currently gaining ground, related to the growing importance of encouraging popular participation, extrapolating the merely biomedical interventions, more classically used.

However, I suggest, since the Introduction, to give more emphasis to the fact that the quality of the articles was not verified. Some conclusions of the original authors are reproduced here, but sometimes there is doubt as to whether good parameters and controls were used. Some examples have been added below.

Thank you, reviewer, #1 for raising this point. Indeed, we did not report and check the quality of the included studies. During the data extraction phase, we also concluded that we had studies that did not use suitable parameters and control groups. However, we did not exclude these studies since our aim was not to include only high-quality studies but to summarise data and report the evidence presented in scientific and grey literature.

The Introduction is adequate and updated, with justified arguments, but it is worth mentioning structural determinants, which go beyond the health sector and which limit the scope of surveillance and entomological control.

Thanks for your remark. You are right; there are other factors worthy of being mentioned that obstruct surveillance and entomological control. We have added the following sentence to the introduction to include your remark. “Also, social-cultural, environmental, and climatic parameters (e.g., urbanisation, building environment, and climate change) limit the scope of surveillance and entomological control.” (See page 4). 

In the Results section, in the item ‘Interventions’, when referring to references 51 and 55, the text says that the use of pyriproxyfen ‘... led to a ten-fold decrease in adult mosquito emergence...’. It is necessary to make it clear here that the difference is the strategy, not the insecticide: this is the effect from the ‘disseminating stations’ (‘Mosquito-Disseminated Insecticide for Citywide Vector Control’).

Many thanks for mentioning this point. We have adjusted the sentence. The following sentence has been added to the manuscript: “One study found that dissemination of pyriproxyfen dust-particles from dissemination stations led to a ten-fold decrease in adult mosquito emergence from sentinel breeding sites.” And the following sentence has been added at the end of the paragraph: “The observed beneficial effect could be due to the type of strategy (mosquito-disseminated insecticide) and not the kind of larvicide used.” (See page 14). 

Still in the item ‘Interventions’, there are issues that are certainly not addressed in the analysis in general review studies of this type. When mentioning, for example, reference 91, which deals with the application of different larvicides, the stability and persistence of Bti formulations in the field, and the resistance of the vector population to insecticides, notably temephos, would be relevant parameters for the analysis of the results.

Thank you for raising this point. Indeed, we did not mention the effect of temephos properly. Therefore, a new sentence that addresses the impact of temephos (presented in the study labelled reference 92, previously 91) has been added to the manuscript. “The usage of temephos granules resulted in 4 weeks of complete control.” (See page 14).

Regarding the last paragraph of the item ‘Interventions’, concluding about the material presented, I reinforce the importance of mentioning the disseminating stations, in this specific case, so that it does not seem that pyriproxyfen is more efficient than the other larvicides presented. The aspect at issue in articles 51 and 55 is not simply the product, but the way of disseminating it in the treated area.

Thanks for highlighting this point. We have incorporated your remark in the last paragraph of the section “Application of larvicide”. The following sentence has been added to the manuscript: “However, it is essential to mention that the observed positive effect of pyriproxyfen is possibly caused by the method of disseminating the larvicide and not the larvicide itself.” (See page 15). We hope we have addressed this point correctly. 

Regarding the item 'Health education and community mobilization/participation' - considering that this control component has increasing relevance, a comment on the formalization, or not, of information on Aedes aegypti in schools in different countries would be worth mentioning. Perhaps in grey literature... Did the authors find something relevant, formal, in this sense?

Thank you for raising this point. Indeed, health education and community mobilisation/participation are essential components of mosquito control. Some studies indicated that it is hard to maintain program implementation and effect when national and regional regulations do not incorporate mosquito control/education. The health system needs to formalise the intervention through regulations to achieve sustainability. The following sentence has been added to the section "health education and community mobilisation/participation:" Furthermore, the government, including the health systems, must formalise mosquito control programs and health education campaigns through regulations to support interventions. Without regulations, it is hard to maintain program implementation." (See page 23).

About Reference 70, the text says that One study that assessed the impact of mass-media communication campaigns did not show beneficial results [70].' It seems a somewhat superficial conclusion, taking into account that, as mentioned in the Manuscript, the quality of articles has not been verified. In this specific case, what was the potential for publicizing the campaigns? Was it verified, for example, if the disclosure actually reached the affected people?

Thank you for mentioning this issue. The cited study aimed to evaluate the coverage and reach of an intervention based on mass-media communication of dengue surveillance reports and its effect on the presence of intra-domiciliary breeding sites for Aedes in Guadalajara de Buga, Colombia. An observational study was performed among 1426 households to identify breeding sites and intervention exposure. A case control study was conducted to determine the intervention’s effect. The study concluded that the intervention was not effective, but the authors also stated that it is important to evaluate the intervention to determine factors that reduced the impact of the strategy. Many factors could have reduced the effect of the intervention, including the method of implementation, distribution of information etc. 

We have adjusted the sentence based on your comment. The following sentence has been added to the manuscript: “One study that assessed the impact of mass-media communication campaigns indicated that although intervention coverage was adequate (59,4% of the population), the reach (people that could recall the content of the information) was low (22,3%). Also, no association between the intervention and the presence of breeding sites was found.” (See pages 21-22). We hope we have addressed your remark properly. 

Still referring to the item ‘Health education and community mobilization/participation’, a comment: the number of articles from Cuba (121-124, 126-128, 130, 131) calls attention. The importance of popular participation in dengue prevention by means of Aedes control in this country is, in fact, already consolidated...

In fact, the importance of Cuba in relation to prevention actions, in general, is reiterated in the item 'Resources and capacity', in reference 123 (Another study indicated that providing adequate basic utility services (e.g., garbage collection) to the communities could have a major bearing on the sustainability of the community-based mosquito control intervention, facilitating the successful implementation of the Aedes control program [123].'). These papers signal not only the stimulus to popular participation, but also the counterpart of public management – an aspect that would deserve to be highlighted.

Thanks for your remark. Indeed, public management is a critical factor that can improve or obstruct public health interventions. We have added a sentence to the manuscript to pay more attention to the mentioned factor. See page 26: “Proper public management (e.g., development and approval of health regulations and public health interventions, intersectoral coordination and creation of linkages between organisations to create a platform to negotiate solutions) is an essential factor that is required to achieve sustainable results”. Hopefully, we have addressed your remark correctly.

In some cases, throughout the manuscript, anecdotal reports are generalized, as if they were related to an entire country; it would be usefull to verify throughout the text. For example, in the item 'Integrated interventions', about reference 45: Another study that evaluated Brazil’s national dengue control plan attributed the program’s lack of success to insufficient household coverage [45].', it would be important to mention here that the article deals with the work in one municipality only.

Thank you for raising this point. We have reviewed the manuscript and made the needed adjustments to address the issue you mentioned. See page: 23. 

Regarding reference 54: The incorporation of the community mobilisation concept into the Brazilian national plan was linked with the reduction of the total number of dengue cases between the years 2009 and 2010 [54].' - there is a difference between the title in Portuguese and English in the original article. The article in Portuguese refers to only one municipality, Ipatinga. In table S3, it corresponds to article 24. The table is ok, it alludes to Ipatinga. But the main text is confusing, there is room for the interpretation that the result obtained is national in scope.

Thank you for mentioning this point. We have made the needed adjustments to specify the scope of the study. The following sentence has been added to the manuscript: “The incorporation of the community mobilisation concept into the Brazilian national plan was linked with the reduction of the total number of dengue cases between 2009 and 2010 in Ipatinga city. “See page 23.

Still in the item 'Integrated interventions', when there is mention of a study in Mexico (reference 93), the statement Dengue incidence decreased from 81.4% in 2010 to 79.1% in 2011 [93].', seems to be a vague conclusion: one point is that differences are too small; moreover, the fact that dengue does not occur homogeneously between years is not being taken into account. For example, the text in table S3 (it is article 63) says that: ‘The interventions reduced the peak of cases that had been recorded in the rainy season resulting from the transmission of dengue’ - that is, it seems that the text of the present Manuscript assumes that (a) there is a significant reduction in the number of cases in the period and that (b) this reduction is a function of the intervention. However, it is known that there are many other factors that may be contributing to this scenario, which could be at least cited, so as not to appear that this conclusion is being shared here.

Thank you for this remark. We know that other factors can influence the results of the study mentioned above and other studies. We do not assume that there is a significant reduction. We reported the findings/conclusions of the study. The study’s results were compared with the number of dengue cases before the intervention and reported dengue cases in other municipalities in Mexico. To reduce confusion, we have removed the sentence “Dengue incidence decreased from 81.4% in 2010 to 79.1% in 2011”. The following sentence has been added to the manuscript: “Two years of implementing an integrated intervention reduced the peak of dengue cases recorded in the rainy season in Colima (Mexico). However, more research is needed to determine the actual effect of the intervention since different factors (e.g., amount of rainfall) can influence the outcome of this study. (See page 24). We hope we have addressed your concerns correctly.

The first mention of the possibility of insecticide inefficiency is in the final, concluding paragraph of the item ‘Integrated interventions’. It is worth dealing with this aspect, in a generic way, since the introduction, explaining that this variable exists, but it was not addressed in the bibliographic survey, neither taken into account in the analyses.

Thank you for mentioning this point. Indeed, insecticide inefficiency is a crucial component that needs to be mentioned when addressing mosquito control interventions. Since the main aim of the scoping review is to document the interventions performed in the LAC region, we paid attention to the mentioned component in the introduction, summary sections and discussion. We tried not to repeat information since the scoping review was already long. Insecticide resistance was addressed on pages 25, 26, 27 and 30. 

In the Discussion, in relation to the statement However, many factors such as insufficient funding, resources, workforce, and political priorities (e.g., prioritising SARS-CoV-2 over ABIDs) obstruct the implementation and maintenance of the IVM program nowadays.' – about 'SARS-CoV-2 over ABIDs': what do you mean 'political priority'? That was a contingency. It is not a question here of opposing one disease to another, but of ensuring a more comprehensive view of health. Finally, it is important to take into account that the focus is not the mosquito.

Thank you for raising this point. What we mean here is that due to COVID, the focus of the government/politics is currently not on mosquito-borne diseases. There is a change in priority to address the challenges that COVID brought. To address your remark, we made some adjustments to the sentence. We have removed the words "over ABIDs". The following sentence has been added to the manuscript: "Besides, the focus of many health systems is not on mosquito control since many countries face a range of more immediate problems." (See pages 29-30). 

The Discussion refers to article 149 to exemplify resistance to insecticides as a factor in reducing the effectiveness of the interventions. This article corresponds to reference 29 of the list in table S4. However, there is no mention of resistance in the table, nor in the abstract of the article. In practice, the only occasion this article, 149, deals with resistance is in its Discussion section, when it cites two other references, both from 1998.

Thank you for your remark. Indeed, the study itself did not assess insecticide resistance in Trinidad but used other studies that managed to test for insecticide resistance in the same study site and time frame to explain the findings. The following sentence can be found in the discussion section of the paper with reference number 149 (in the new version of our manuscript, reference 150): "However, studies on the Aedes aegypti insecticide susceptibility profiles conducted during this period have shown resistance to organophosphates and carbamates in Trinidad". We have reviewed the included studies that mentioned the possible implications of insecticide resistance. They did not assess insecticide resistance but used available literature and reported it in the discussion section. We have added more references of studies that reported insecticide resistance. (See page 30). We hope we have addressed your remark correctly.

In the Conclusions, the manuscript states that The integrated Aedes control approach was more effective than a single strategy.'. Considering that there was no evaluation of data quality of the articles, as explained in the paragraph before this one in the main text (‘limitations and strengths’), it is not clear how this conclusion was reached. It is quite true that this conclusion is, at least intuitively, desired. But the parameters evaluated here do not allow such conclusions. On the other hand, it seems relevant (and could be emphasized) that this work contributes to systematize, classify, or quantify the different methodologies approached; it would be possible, for example, to present, even graphically, the relative contribution of different countries, or regions, to each type of approach. Another option is to make it clear that the higher effectiveness of an integrated approach over single strategies was the major conclusion of the analyzed articles - and not of the authors of this manuscript.

Thank you for mentioning this point. The conclusion is based on synthesised information, not our understanding/beliefs. We adjusted the conclusion section to make it clear that the conclusion on the integrated approach is based on the findings of the studies we included. The following sentence was added to the manuscript: “The major conclusion of the synthesised data was the higher effectiveness of the integrated Aedes control approach over single strategies.” (See pages 2 and 31).

In the Acknowledgments, the 'thank you in advance' calls attention: We want to thank both the editors and reviewers for their insightful comments on the article.'

Thank you for raising this point. We intended to thank all the reviewers since they invested time in reading and reviewing our manuscript during the review phase. Once again, we appreciate your comments even though the paper is not accepted. Each feedback helps us to increase the quality of our manuscript.

Minor comment: I did not systematically check the bibliographic references, in relation to questions of form. However, I noticed that reference 34 (Camargo Donalisio MR, Leite OF, Mayo RC, Pinheiro Alves MJC, de Souza A, Rangel O, et al. Use of Temephos for Control of Field Population of Aedes aegypti in Americana Sao Paulo, Brazil. 2002.') is incomplete. It remains to add: ‘Dengue Bulletin. 2002 Dec; 26: 173-177'.

Thanks for mentioning this point. We have reviewed the reference list and made needed adjustments.

Reviewer #2: Is a well written and important scoping review, my congratulations to the authors. This review give valuable information for control programas in Latin America and the Caribbean. In tha attached file, the authors can found somo suggestions.

Thank you, reviewer, #2. We have adjusted the manuscript based on your comments. See manuscript with track changes.

---

## [Editor Report · Decision Letter 1]

19 Oct 2022

Effectiveness of Aedes-borne infectious disease control in Latin America and the Caribbean region: a scoping review

PONE-D-22-16967R1

Dear Dr. Mulderij-Jansen

We’re pleased to inform you that your manuscript has been judged scientifically suitable for publication and will be formally accepted for publication once it meets all outstanding technical requirements.

Kind regards,

Rafael Maciel-de-Freitas

Academic Editor

PLOS ONE

---

## [Editor Report · Acceptance letter]

24 Oct 2022

PONE-D-22-16967R1 

Effectiveness of *Aedes*-borne infectious disease control in Latin America and the Caribbean region: a scoping review 

Dear Dr. Mulderij-Jansen:

I'm pleased to inform you that your manuscript has been deemed suitable for publication in PLOS ONE. Congratulations! Your manuscript is now with our production department. 

Kind regards, 

on behalf of

Dr. Rafael Maciel-de-Freitas 

Academic Editor

PLOS ONE